# Biogenic Sulfide-Mediated Iron Reduction at Low pH [note 1]

**DOI:** 10.3390/microorganisms12101939

**Published:** 2024-09-25

**Authors:** Caryl Ann Becerra, Brendan Murphy, Brittnee V. Veldman, Klaus Nüsslein

**Affiliations:** 1Department of Biology, California State University Channel Islands, Camarillo, CA 93012, USA; 2Cambrian BioPharma, Inc., New York, NY 10003, USA; brendanmurphy837@gmail.com; 3Department of Chemistry, California State University Channel Islands, Camarillo, CA 93012, USA; brittnee.veldman@csuci.edu; 4Department of Microbiology, University of Massachusetts Amherst, Amherst, MA 01003, USA; nusslein@microbio.umass.edu

**Keywords:** attenuation of acid mine drainage (AMD), sulfate-reducing bacteria (SRB), iron reduction, sulfate reduction, biogenic sulfide, microcosms

## Abstract

Acid mine drainage (AMD) pollutes natural waters, but some impacted systems show natural attenuation. We sought to identify the biogeochemical mechanisms responsible for the natural attenuation of AMD. We hypothesized that biogenic sulfide-mediated iron reduction is one mechanism and tested this in an experimental model system. We found sulfate reduction occurred under acidic conditions and identified a suite of sulfate-reducing bacteria (SRB) belonging to the groups *Desulfotomaculum*, *Desulfobacter*, *Desulfovibrio*, and *Desulfobulbus*. Iron reduction was not detected in microcosms when iron-reducing bacteria or SRB were selectively inhibited. SRB also did not reduce iron enzymatically. Rather, the biogenic sulfide produced by SRB was found to be responsible for the reduction of iron at low pH. Addition of organic substrates and nutrients stimulated iron reduction and increased the pH. X-ray diffraction and an electron microprobe analysis revealed that the polycrystalline, black precipitate from SRB bioactive samples exhibited a greater diversity of iron chalcogenide minerals with reduced iron oxidation states, and minerals incorporating multiple metals compared to abiotic controls. The implication of this study is that iron reduction mediated by biogenic sulfide may be more significant than previously thought in acidic environments. This study not only describes an additional mechanism by which SRB attenuate AMD, which has practical implications for AMD-impacted sites, but also provides a link between the biogeochemical cycling of iron and sulfur.

## 1. Introduction

Acid mine drainage (AMD) is a form of environmental contamination detrimental to aquatic life and water quality. It is produced from mining activities that expose sulfide-rich minerals such as pyrite to chemical and biological oxidation. Oxidation of pyrite produces sulfuric acid, which leads to the dissolution of metals such as iron [1]. The resulting acidic water with higher concentrations of dissolved metals is unsuitable for aquatic and wildlife, which necessitates the investigation of remediation strategies for AMD.

Our study site, Davis Mine in Rowe, Massachusetts, is an AMD site. However, unlike in other AMD systems, here, the environmental contamination is naturally attenuated within 200 m downstream of the mineshaft where it originates. Surface and groundwater sampled during a two-year period revealed that the contamination is contained within a low-drainage area in contact with waste rock piles and groundwater [2]. These characteristics make this area a model study site to explore the mechanisms that have led to the natural attenuation of AMD. Understanding how this natural attenuation occurs may offer strategies for actively remediating AMD.

Microbial communities play a crucial role in the biogeochemical mechanisms that contribute to the natural attenuation of AMD [3]. At Davis Mine, evidence of biological processes that could lead to the attenuation of AMD has been found at Davis Mine. For example, if sulfate reduction is occurring, then the pH of the water should increase while the oxidation–reduction potential and the quantity of dissolved metals decrease [4,5,6,7]. In fact, the concentration of dissolved metals gradually decreases downstream toward the periphery of the contaminated site where near-neutral pH and lower sulfate concentrations in the groundwater have been measured [8]. SRB reduce sulfate to sulfide. Under acidic conditions, this sulfide-ion forms hydrogen sulfide. At Davis Mine, we observed that sampled groundwater from wells smelled of hydrogen sulfide. Additionally, when sulfide binds with dissolved metals to form an often black precipitate, the sulfide is either transported or settled out of the water column. As a result, the co-precipitation of the metal sulfide further alleviates the environmental problem—high concentrations of dissolved metals [9,10,11,12]. Illustrating this process, at the subsurface of Davis Mine, we found black precipitates suspected to be iron sulfide in the subsurface layers of Davis Mine. These field observations support the idea that sulfate-reducing bacteria are contributing to the attenuation of AMD at Davis Mine.

Laboratory microcosms with sediment and groundwater of the site showed similar effects of a biologically influenced AMD attenuation as the effluent stream. When the microcosms were amended with organic carbon such as algae extract or glycerol, or when the incubation temperature was raised, the effects mentioned above were enhanced [13]. Both field and laboratory microcosms suggested a biological component to the attenuation of AMD, and this study investigated the role of SRB in this attenuation.

The specific mechanisms in which SRB play a role in attenuating AMD at this site had yet to be determined. For example, it is known that SRB can reduce sulfate to sulfide. However, some SRB also enzymatically reduce iron, which could also contribute to AMD attenuation [14,15,16,17]. Another possible attenuation mechanism is the nonenzymatic reduction of iron via sulfide produced by SRB. It has been previously shown that sulfide will reduce iron; however, this has not been demonstrated under natural in situ acidic conditions like our study site [18,19,20,21].

In this study, we tested for the following scenarios: If SRB can enzymatically reduce iron, then (i) stimulating SRB will result in an increase in Fe^2+^ and (ii) Fe^2+^ will still be produced even when iron-reducing bacteria are inhibited, and SRB are inhibited from reducing sulfate. If SRB do not enzymatically reduce iron, then (i) the Fe^2+^ will be produced abiotically through reaction with biogenic sulfide; (ii) increasing the rate of sulfate reduction will result in an increase in Fe^2+^ production; and (iii) iron sulfide precipitate will form.

## 2. Materials and Methods

### 2.1. Study Site Description

Established in 1882, Davis Mine in Rowe, Massachusetts was once the largest working pyrite mine in the state (Figure 1). After its collapse in 1909, the mine filled with groundwater that exited continuously from the old mine shafts and flowed over roughly 3 hectares of waste rock tailings piles, generating AMD. Today, it is a single AMD effluent stream that runs through Davis Mine and joins Davis Mine brook [22]. The mine area exhibits a patchiness of regions that display higher acidity and concentrations of sulfate and dissolved metals intermixed with regions of reduced acidity and lower concentrations of sulfate and dissolved metals [8].

### 2.2. Sample Collection

At Davis Mine, sediment samples were collected within an area where AMD was attenuated that was approximately 25 m from a well that had AMD and approximately 200 m from the mine shaft where the AMD effluent stream originated (42°40′51.3″ N 72°51′50.3″ W). This location was chosen because it was the most attenuated of all of the wells studied in our group in this area [2,8,13]. Sediment cores were taken in July 2007 at a depth of 40 to 56 cm below the surface. At this depth, indications of sulfate reduction such as black precipitates with an accompanying smell of hydrogen sulfide were observed. Sediment cores of 15 cm in length and 5.0 cm in diameter were collected using a sediment corer with a slide hammer attachment. Groundwater was collected from an adjacent borehole well at approximately 0.5 m from the extracted sediment cores of an AMD-attenuated area. Immediately after sampling, the sediment cores were kept in N_2_ (g)-filled mason jars and on ice during the one-hour transport back to the laboratory.

### 2.3. Microcosms Preparation and Analysis

A slurry consisting of a 1:1 ratio of sediment to autoclaved groundwater was prepared in an Ar (g)-filled anoxic chamber. Each microcosm was filled with 10 mL of sediment slurry and 90 mL of autoclaved groundwater, and with the remaining headspace filled with Ar (g). Crimp sealed microcosms were incubated in the dark without agitation at a constant temperature of 20 °C. Three types of amended microcosms were constructed. Carbon- and nutrient-amended microcosms, which are glycerol (G), nitrogen (N), and phosphate (P), denoted GNP-amended for 84 mg/L of glycerol, 30 mg/L of ammonium sulfate, (NH_4_)_2_SO_4_, and 31 mg/L KH_2_PO_4_, were created as previously described [13]. A set of microcosms was amended with 5 mM molybdate to selectively inhibit SRB, and another set was amended with 20 mM nitrate to competitively inhibit both SRB and FeRB.

To account for any changes due to abiogenic geochemical processes, abiotic controls were created by autoclaving microcosms 3 times on consecutive days to inactivate any biological agents including spores, essentially making these microcosms sterile.

To account for the effect of sulfide on Fe^2+^ production in an AMD system, as well as extent of natural precipitation or background redox reactions through the interaction of sulfide, we created a set of sterile microcosms (abiotic controls) that also contained 0.1 mM and 1.0 mM sodium sulfide.

To evaluate the composition of the black precipitates formed in the presence of SRB without soil contamination, soil-free Davis Mine water samples filtered using 0.45 µm pore size (Millipore, Billerica, MA, USA) were used for mineral and elemental analysis. Two of the soil-free samples were prepared following the abiotic control microcosms autoclaving procedure. To each of these, elemental abiotic control samples were spiked with sodium sulfide to a final concentration of 5 g/L of Na_2_S, which was added along with either 5 g/L Fe^2+^ chloride or 5 g/L Fe^3+^ chloride. A third elemental control sample of the bioactive mine water was also spiked with sodium sulfide at 5 g/L of Na_2_S. All three samples produced black precipitates shortly after the addition of sulfide and iron salts but were allowed to rest without agitation at a constant temperature of 20 °C. All precipitates were concentrated by centrifugation at 50,000× *g* for 30 min, then collected by filtration onto nitrate cellulose membrane filters with 0.8 µm pore size (Millipore, Billerica, MA, USA) in an anoxic chamber, where the filter was left to dry overnight at room temperature. Samples were stored under N_2_ (g) until analysis by X-ray diffraction (XRD) and electron microprobe analysis.

Homogenous microcosm samples were extracted aseptically using sterile syringes on the day of inoculation and periodically during the course of the incubation period. The pH and oxidation–reduction potential (ORP) values were measured using a pH/ORP meter (Thermo Orion, Waltham, MA, USA). Ferrous ion concentration was measured spectrophotometrically (Thermo Spectronic Genesys 5, Pittsburg, PA, USA) following the ferrozine assay [23,24]. Dissolved organic carbon (DOC) was measured using a Shimadzu TOC-VCPN analyzer with ASI-V autosampler (Shimadzu Corporation, Kyoto, Japan).

### 2.4. Characterization of Precipitate

Powder X-ray diffraction was performed with a mortar and pestle, fine-powdered samples using a PANalytical X’Pert Pro diffractometer using the X’Celerator Module (PANalytical, Westborough, MA, USA), and copper tube (PW3373/10 Cu LFF DK184016) with theta–theta goniometer (240 mm radius) generating Cu Kα radiation (λ = 0.154 nm). The samples were mounted on a zero-background silicon sample holder at room temperature and step-scanned using 45 kV and 40 mA at a 2θ range of 5° to 75° with 0.017° 2θ/s steps. Scans were made using a divergence slit size of 0.0290°, a specimen length of 10.00 mm, and a receiving slit size of 0.100 mm. Diffraction patterns of all samples were processed and overlaid using X’Pert HighScore (https://www.malvernpanalytical.com/en/products/category/software/x-ray-diffraction-software/highscore) and X’Pert Data View (https://www.malvernpanalytical.com/en/products/category/software/x-ray-diffraction-software/data-collector) (PANalytical, Westborough, MA, USA) [25].

Elemental composition analysis was performed on specimens using a CAMECA Ultrachron SX50 electron microprobe with SXRayN50 software (https://www.cameca.com/service/software/peaksight) and PAP matrix correction (Cameca, Mahwah, NJ, USA). This instrument has four wavelength-dispersive spectrometers with P10 β low-proportional counters: two counters at approximately 1 bar of pressure and two counters at 3 bars of pressure. For the analysis of YLγ, ThMα, PbMα, and UMβ, we used TAP (low P), PET (low P), PET (high P), and PET (high P), respectively. Samples of eight to ten randomly selected grains of precipitate were mounted on a glass slide as a standard petrographic thin section embedded in epoxy resin. To prevent absorption of current fluctuation and beam damage, a carbon coat (250 Å thickness) was applied to the outside of each mount [26].

The groundwater used in all samples was collected from a single site at the Davis Mine as labeled in Figure 1, and was found to contain pH 5.3 ± 0.2; ORP 253 ± 33; total ionic iron 46 ± 10 ppm, Mn^2+^ 2.3 ± 0.1 ppm, Zn^2+^ 1.4 ± 0.6 ppm, Cu^2+^ 83 ± 50 ppb, Al^3+^ 4.0 ± 2.4 ppb, Pb^2+^ 30 ± 4.3 ppb, Na^+^ 2.6 ± 0.1 ppm, Mg^2+^ 18 ± 1 ppm, K^+^ 85 ± 32 ppm, Ca^2+^ 117 ± 6 ppm, Si^2+^ 13 ± 1 ppm, Cl^−^ 9.3 ± 1.2 ppm, F^−^ 0 ± 0 ppb, Br^−^ 0 ± 0 ppb, NO_3_^−^ 32 ± 55 ppb, PO_4_^−^ 0 ± 0 ppb, and SO_4_^2−^ 608 ± 52 ppm. Cations were determined via ICP-OES on a Spectro^®^ M120 sequential instrument (Kleve, Germany). Anions were determined on a Lachat^®^ IC5000 ion chromatograph (Milwaukee, WI, USA) [2].

### 2.5. Sulfate Reduction Rate (SRR)

To determine the sulfate reduction rates, the unamended microcosms, GNP-amended microcosms, and abiotic controls after 20 days of incubation were injected with 3.0 µL of radiolabeled sodium sulfate, ^35^SO_4_^2−^ (Amersham Biosciences, Piscataway, NJ, USA) and incubated in the dark at 18 °C. Sulfate reduction was stopped by adding 20% (*w*/*v*) zinc acetate to the sample and kept at −20 °C. The aqueous and the sediment fractions of the microcosms were separated, and the sediment washed to remove radio-labeled sulfate that was not taken up by SRB. The washed sediment was added to a modified one-step distillation to which 20 mL of dimethylformamide, 8 mL of 6 N HCl, and 16 mL of 1 M CrCl_2_ was injected anoxically. The mixture was bubbled with N_2_ (g) for 2 h. The liberated sulfide was precipitated as ZnS in the zinc acetate solution trap. The radioactivity was measured on a scintillation counter as counts per minute (cpm). The rate of sulfate reduction was determined by comparing the activity of the total reduced sulfur to the activity of the total radiolabeled sodium sulfate [27].

### 2.6. Quantification of Sulfate-Reducing Bacteria (SRB)

After DNA extraction using the UltraClean Soil DNA isolation kit (MO BIO Laboratories, Solana Beach, CA, USA), the quantities of SRB in unamended microcosm and GNP-amended microcosms were determined in triplicate by quantitative PCR of the *dsr* gene using primers DSR1F and DSRQ2R [28]. Each reaction contained 10 µL 2X SYBR Green master mix (Stratagene, La Jolla, CA, USA), 0.3 µL of 1:500 Rox (Strategene, La Jolla, CA, USA), 2.4 µL of each primer to a final concentration of 250 nM, 3.3 µL of H_2_O, and 1.6 µL of template. The same concentration of template for each sample was used as determined with a spectrophotometer (Nanodrop1000, Wilmington, DE, USA). Using a qPCR thermocycler MX3005P (Stratagene, La Jolla, CA, USA), the reactions were run as follows: 40 cycles of 95 °C for 30 s, 55 °C for 1 min, 72 °C for 1 min with an initial denaturation step at 95 °C for 10 min, and a final extension at 55 °C for 30 s. Fluorescence was measured at the end of each cycle. Amplicons of *dsr* were diluted and used as standards for qPCR to determine the quantity of *dsr* in each microcosm. Three standard curves with R^2^ > 0.999 and efficiency above 83% were constructed.

Reaction mixtures for PCR amplification of 16S rRNA genes of six specific SRB groups were prepared with 1X buffer; 0.25 mM each of dNTP, 0.08 U/10 µL Taq polymerase (all from Promega, Madison, WI, USA), 2.0 mM MgCl_2_, and 400 ng/ µL BSA (Sigma, St. Louis, MO, USA); 0.5 µM each of primer sets DFM140 and DFM842, DSB127 and DSB1273, DBB121 and DBB1237, DBM169 and DBM1273, DCC305 and DCC1165, and DSV230 and DSV838 (Integrated DNA Technologies, Coralville, IA, USA); and 3.76 ng/µL DNA template in a final volume of 30 µL [29]. Amplification of the SRB group-specific 16S rRNA genes were performed in a PTC-200 Peltier thermal cycler (MJ Research, Waltham, MA, USA). Reactions were carried out in 30 cycles as follows for the amplification of the SRB group-specific 16S rRNA genes: 94 °C for 30 s, 55 °C for 30 s, 72 °C for 90 s with an initial denaturation step at 94 °C for 5 min, and a final extension at 72 °C 5 min with the exception of the annealing temperatures as follows: DFM set at 58 °C, DBB set at 66 °C, DBM set at 64 °C, DCC set at 65 °C, and DSV set at 61 °C [29]. Amplicons were purified prior to ligation using QIAquick PCR purification kit (Qiagen, Valencia, CA, USA). PCR products were confirmed by agarose gel electrophoresis and measuring the concentration using a nanodrop spectrophotometer (ND1000, Wilmington, DE, USA).

### 2.7. DAPI Staining for Total Cell Count

For a total direct cell count based on the nucleic acid stain 4′,6-diamidino-2-phenylindole (DAPI), the samples were fixed in a 3:1 volume of 4% paraformaldehyde and incubated overnight at 4 °C following manufacturer’s protocol (Thermo Fisher Scientific, Waltham, MA, USA). Samples were then concentrated by centrifugation, washed in 1X phosphate saline buffer (0.9 g of NaCl, sodium phosphate buffer 15 mM, pH 7.2), and concentrated again by centrifugation at 16,000× *g* for 10 min. The sediment slurry was sonicated for 30 s with Tween-20 (0.01% final concentration) and diluted 1:150 with phosphate saline buffer. The diluted sample was filtered through an MCE, Fisherbrand membrane (0.2 μm pore size, 25 mm in diameter, Fisher Scientific, Pittsburg, PA, USA). DAPI in an antifade reagent was added directly to the membrane (ProLong Gold, Invitrogen, Carlsbad, CA, USA) and allowed to cure overnight at room temperature. The membranes were viewed under an epifluorescence microscope (Nikon Eclipse E400, Tokyo, Japan) with a Hamamatsu digital camera (Shizuoka, Japan) and 30 fields per sample were counted.

## 3. Results

### 3.1. Inhibition and Stimulation of Sulfate-Reducing Bacteria (SRB) in Microcosms

To determine the effect of sulfate-reducing bacteria (SRB) on the production of Fe^2+^, the microcosms were amended to either stimulate or inhibit the SRB (Figure 2a). The GNP-stimulated SRB (0.67 ± 0.12 mM, day 1, 8.89 ± 1.46 mM, day 46, n = 3) showed a significant increase (T 95%) in Fe^2+^ production over the unamended microcosms (0.50 ± 0.46 mM, day 1, 2.35 ± 0.42 mM, day 46, n = 3) over a 46-day incubation. In contrast, no detectable Fe^2+^ was produced in microcosms that were amended with molybdate, a selective inhibitor for SRB, or nitrate, a competitive electron acceptor. The abiotic control exhibited an increase in Fe^2+^ from 1.32 ± 0.19 mM to 2.50 ± 0.44 mM, (n = 3) throughout the course of the incubation.

An increase in pH suggests that AMD attenuation occurred and therefore was monitored throughout the incubation period as well (Figure 2b). The pH of both the GNP-stimulated and unamended microcosms exhibited a significant increase (T95%) in pH from 3.37 ± 0.05 mM to 4.39 ± 0.03 mM, and from 3.06 ± 0.06 mM to 3.81 ± 0.10 mM (n = 3), respectively, throughout the course of the incubation. The pH of the abiotic control was comparable to the molybdate-amended microcosms with a pH average of 3.5, with no significant change during incubation.

Another indication that SRB were active was the development of black precipitates. Black precipitates developed in unamended microcosms and in GNP-amended microcosms. Also, while sampling these microcosms the odor of hydrogen sulfide, an indication of sulfate reduction was noticed.

### 3.2. Sulfate Reduction Rates

The sulfate reduction rates of a second set of unamended microcosms, GNP-amended microcosms, and abiotic controls were determined (Table 1). The sulfate reduction rates of the unamended microcosms had a mean of 19 ± 2 nmol/(cm^3^d). The sulfate reduction rates of the GNP-amended microcosms were 22 ± 2 nmol/(cm^3^d). At 95% confidence level using the *T*-test, there was no significant difference in the sulfate reduction rates of both sets of microcosms. The rates for the corresponding controls were negligible at 0.31 and 0.29 nmol/(cm^3^d), respectively.

In addition to sulfate reduction rates, we also monitored the pH, ORP, DOC, sulfate, and Fe^2+^. The increase in pH over time was higher and significantly different in the GNP, unamended, and ground water samples compared to the abiotic control and corresponded to the decrease in ORP. The pH of the abiotic controls remained relatively constant; however, the ORP increased, indicating oxidation. The pH and ORP of the pore water extracted from the sediment core that was used to construct the microcosms were like the pH and ORP of the microcosms at the end of the incubation. Negative ORP values indicated that reducing conditions were achieved. The greatest decrease was seen in the GNP-amended microcosms, followed by the unamended microcosms and the abiotic controls. The average sulfate concentrations for the unamended and GNP-amended microcosms were both higher than their respective abiotic controls. All ORP values were statistically unique to 95% confidence.

### 3.3. Abundance and Diversity of SRB

Enumeration of SRB was based on the quantification of the functional gene *dsr*, which encodes the dissimilatory sulfite reductase [28]. At the start of the incubation (t = 0), both the unamended and GNP-amended microcosms had similar counts of SRB, but by the end of the incubation period, the number of SRB in the GNP-amended microcosms had more than doubled the number of SRB seen in the unamended microcosms (Table 2).

Based on the DAPI counts of the total cell number, the percentage of SRB in the total population increased over time. The relative abundance of SRB in the unamended microcosms did not show a statistically significant change during the incubation period, where the GNP-amended did show a significant difference to a 95% confidence. With the SRB group-specific primers, we detected more groups of SRB in the GNP microcosms than the unamended microcosms (Table 3). SRB groups *Desulfotomaculum*, *Desulfovibrio*, and *Desulfobulbus* were found in both GNP microcosms than the unamended microcosms with the addition of *Desulfobacter* in the GNP microcosms.

### 3.4. Origin and Characterization of Precipitate

Black precipitates developed immediately in the abiotic controls after the addition of sulfide, which accompanied an increase in Fe^2+^ concentration and pH (Figure 3a,b). The amount of Fe^2+^ produced corresponded to the amount of sulfide added: the average concentration of triplicate samples of Fe^2+^ generated in the presence of 0.1 mM and 1.0 mM sodium sulfide was 0.14 ± 0.2 mM and 0.98 ± 0.4 mM, respectively (Figure 3a). Similarly, the change in pH value also corresponded to the amount of sulfide added (Figure 3b). During the incubation, the pH of the 0.1 mM and 1.0 mM sodium sulfide-amended microcosms was approximately 2.8 and 3.4, respectively. Therefore, Fe^2+^ production and increase in pH were correlated.

The black precipitates that developed in the abiotic soil-free samples and the GNP-amended microcosms were analyzed using X-ray diffraction to determine the structure and identity of the precipitates (Figure 4) as well as an electron microprobe to determine the elemental composition (Table 4). The precipitates from the three soil-free samples were fine-grained powders that exhibited diffractograms with broad peak widths indicative of previously reported samples generated under similar conditions [30]. Peak widths are directly related to crystallite size, with precipitate formation being observed.

Shortly after initiation in addition to co-precipitation of several minerals, broader peaks are to be expected [31]. Though the large deviation in the GNP sample excludes statistical comparison, the soil-free abiotic samples are significantly similar and significantly different from the Davis Mine biotic sample to 95% confidence in iron concentration.

All three samples exhibit the same major peaks, which predominantly correspond to higher valent iron chalcogenide minerals such as hematite, magnetite, and greigite. However, the bioactive sample exhibits an additional range of minerals that are noteworthy for having a smaller average oxidation state of iron near or equal to two (troilite and pyrite). In addition, several of the additional peaks in the bioactive sample indicate minerals that adsorb or incorporate other metals from the groundwater (pseudomorphic pyrite and chalcopyrite), as shown in Figure 4 [32,33]. Metals such as copper and zinc were found in the groundwater, as listed previously. Though they do not seem to exist in high enough concentrations to contribute to the abiotic background precipitation, they do appear in the more complex mineralization in the bioactive sample. Average iron valence drops with bioactive samples as well as more mineralized precipitation.

Electron microprobe images of representative grains of the microcosm and soil-free abiotic controls are shown in Figure 5. The black precipitates of the microcosm appeared fine-grained and film-like (Figure 5a), compared to the more granular abiotic samples (Figure 5b,c). This trend is echoed in the elemental analysis with the tightly correlated compositions and smaller deviations than either bioactive sample (Table 4).

## 4. Discussion

Acid mine drainage (AMD) is an environmental contamination caused by mining activities that destroys aquatic ecosystems [1]. Yet, there are sites where the AMD is naturally attenuated [7,13,34,35]. Understanding how AMD is attenuated may offer strategies for actively remediating AMD. Previous studies on the natural attenuation of our AMD study site suggests that SRB are key [2,8,13,25]. SRB have been used in AMD remediation in other studies because they reduce sulfate [10,11,25,27,28,29,30,34,35,36,37,38,39]. However, in this study, we determined that SRB, and not FeRB, are also responsible for iron reduction, which contributes to remediation by co-precipitation of metal contaminants.

### 4.1. Iron Reduction Mediated by Sulfate-Reducing Bacteria

Fe^2+^ ions are generated through enzymatic or abiotic reactions. For example, iron-reducing bacteria (FeRB) and some sulfate-reducing bacteria (SRB) can enzymatically reduce iron, while sulfide as the only environmentally relevant compound can reduce iron abiotically [14,16,18,19,21,40,41,42,43,44]. In a bioreactor study that tested a pH range of 6.0–7.5, sulfate and iron reduction occurred at varying ratios that depended on the pH and the acetate concentration. The researchers of that study found that iron reduction occurred to a greater extent if sulfate reduction was occurring [45]. In this study, we determined the contribution of SRB versus FeRB to the production of Fe^2+^ ions by selectively inhibiting these bacterial groups.

SRB are metabolically inhibited by molybdate, a sulfate analog, while FeRB are not affected [16,46]. Nitrate is a more energetically favorable terminal electron acceptor and therefore was used to competitively inhibit the reduction of both iron and sulfate [47,48,49]. Although some SRB are able to reduce nitrate to nitrite, no increase of Fe^2+^ ion concentrations would be due to this metabolism [50]. In addition, nitrite may also inhibit SRB [51,52]. In our microcosms amended with nitrate, no Fe^2+^ ions were produced. In molybdate-amended microcosms, the Fe^2+^ ion concentration was similar to the nitrate-amended microcosms, which had both SRB and FeRB inhibited. Therefore, it is the acidotolerant SRB that are involved in the production of Fe^2+^ ions.

We recognize the potential role of aerobic iron oxidizers, which could also reduce iron. However, our experimental controls were designed to address this possibility. We defined direct iron reduction as abiotic processes or those not influenced by microbial activity. The absence of significant changes in Fe^2+^ concentration in abiotic and inhibited microcosms supports that direct iron reduction was not a major factor.

Although we did not specifically target iron-oxidizing microorganisms that can shift to iron reduction, we observed no Fe^2+^ production in molybdate- or nitrate-amended microcosms, where SRB or iron-reducing bacteria were inhibited. This lack of Fe^2+^ production indicates that if aerobic iron oxidizers were present, their contribution to iron reduction was below the detection level of our assays. Comparison between microcosms with inhibited SRB and unamended microcosms further confirmed that the observed iron reduction is primarily mediated by SRB.

Previous studies related to AMD remediation have shown that the addition of certain organic carbon stimulated microbial activity [37,38,53,54,55,56]. In earlier investigations and in this study, the addition of GNP as an organic carbon and micronutrient source increased the rate of iron reduction [13]. The choice in organic substrate was based on previous microcosms testing acetate, ethanol, pyruvate, lactate, and glycerol [57] as well as readily available, complex, organic substrates found at Davis Mine such as wood chips, algae [13], and cow manure to test for a potential, large-scale, on-site remediation [58]. An increase in the generation of Fe^2+^ ions as compared to unamended microcosms and abiotic controls indicates that iron reduction is biologically mediated under acidic conditions. We chose to amend the microcosms with a low concentration of glycerol because acidophilic microbes do not tolerate high concentrations of organic carbon [39,59,60]. In a comparison to a study of two AMD-treating bioreactors containing acidophilic sulfidogenic consortia, glycerol was used and consumed at a pH of 2.5 [61]. The results of the inhibition and stimulation experiment indicate that SRB mediate iron reduction.

### 4.2. Abundance and Diversity of SRB

SRB are present in greater abundance and activity in acidic environments than previously thought [62]. In our microcosms, the number of DAPI-stained cells remained constant from beginning to end of the incubation period. However, the number of SRB cells detected by qPCR of the *dsr* gene increased from 10^4^/mL to 10^6^/mL. As a comparison, in an iron-dominated, mining-impacted freshwater sediment, the abundance of cultivable SRB using the most probable number was within the range of 10^4^ to 10^6^ cultivable SRB g^−1^ wet weight sediment [63].

In this study, we tested for the presence of six major phylogenetic groups of SRB based on Daly 2000 [29]. Recently, 13 phylum-level lineages that were not known to have *dsr* genes brought the total to 20 distinct phylum-level lineages currently. However, delta-proteobacteria has the largest number with 34 genomes reported [64]. Therefore, in the context of this study, the six major groups that we investigated were of the delta-proteobacteria and Gram-positive for *Desulfotomaculum*.

The detected SRB groups depended on the length of the incubation and the type of amendments made to the microcosms. For example, using SRB group-specific primers, we detected *Desulfotomaculum*, *Desulfovibrio*, and *Desulfobulbus* in the unamended microcosms. However, using SRB group-specific primers for *Desulfococcus*–*Desulfonema*–*Desulfosarcina* and *Desulfobacterium*, these SRB groups were not detected. For the GNP-amended microcosms, the same groups in the unamended microcosms were detected with the addition of *Desulfobacter*, which suggests low abundance at our site. *Desulfobacter* was only detected after 40 days of incubation and with the addition of GNP. In another AMD study, the authors also did not detect *Desulfovibrio* and *Desulfosporosinus* in their sediments but abundantly in their microcosms that had been amended with hydrogen [65]. *Desulfotomaculum* have been found in low pH environments and *Desulfovibrio*, *Desulfobulbus*, and *Desulfobacter* have been found in other AMD sites [66,67]. *Desulfosporosinus* was detected in a study that compared two acidophilic sulfidogenic consortia in bioreactors that treated AMD as well as in acidic mine pit lakes [59,66]. In a mildly acidic AMD site, the dominant clone sequence, representing 65% of the community, was related to SRB, *Desulfosarcina variabilis* [68].

Sequences related to SRB have also been found in other natural acidic environments such as volcanic-impacted water bodies and hydrothermal springs [12,56,69,70,71]. Fortin et al. recovered SRB in acidic and oxidizing conditions, while Senko et al. isolated an acidotolerant SRB able to reduce Fe(III)(hydr)oxide better under acidic conditions of pH 4.4 than at neutral pH 7.1 [42,59]. The genera that accounted for 100% of the SRB-like populations in a study that tested five types of sulfate-reducing passive bioreactors were *Desulfosporosinus*, *Clostridium*, *Ruminococcus*, *Desulfovibrio*, and *Desulfobulbus* [72]. Therefore, SRB are ubiquitous in nature and engineered systems like bioreactors.

### 4.3. Origin and Characterization of the Sulfide Precipitate

Despite the acidic nature of AMD sites, the formation of black FeS precipitates has previously been observed where sulfate reduction or SRB were detected [12]. In fact, this phenomenon has been used in a bioreactor treating acid rock drainage where zero-valent iron as an electron donor was added to produce metal sulfide precipitates [6]. Based on powder X-ray diffraction and electron microprobe elemental analysis, the polycrystalline, black precipitate from both bioactive and abiotic samples show that the presence of free ions in solution can lead to some precipitation or higher valence iron chalcogenide minerals. However, the diversity of minerals, minerals with reduced iron oxidation states, and minerals incorporating multiple metals were significantly increased in the bioactive samples. Black FeS precipitates have been detected when Na_2_S was added to liquid medium or to sulfate-reducing bacterial cultures in controlled environments [20,30]. For example, Gramp et al. detected iron sulfide minerals, greigite, mackinawite, and pyrite forming in media containing SRB [30]. At our study site, iron minerals such as goethite, pyrrhotite, chalcopyrite, pyrite, and magnetite were previously detected [2]. In aqueous samples from our microcosms, we observed that the production of Fe^2+^ ions was directly correlated with increasing sulfide concentration, which agrees with other studies [9,20,73,74]. Previous studies have shown that sulfide reduces iron oxy(hydr)oxides minerals such as lepidocrocite, goethite, magnetite, and hematite [7,41]. Although not a mineral, in another study that used SRB-immobilized particles to treat simulated AMD, “sulfate green rust”, another type of iron sulfate precipitate, developed in their system [75]. More examples demonstrating the iron sulfide sorption of other metals include mercury [76] and copper [77], as seen in the bioactive sample in Figure 4.

Another implication of this study is the improved sequestration of iron and other dissolved metals in the presence of SRB. Dissolved metals may precipitate as metal sulfides. For example, galena is formed through sulfide co-precipitation produced by SRB [56]. Similarly, iron can precipitate as a ferrous sulfide [9,54,59]. Another example is the sulfidation of arsenic on ferrihydrite to arsenic trisulfide in groundwater. Additionally, Fe^2+^ and sulfide could form mackinawite, onto which arsenate would adsorb [78]. In a study of mine-impacted freshwater, the scanning electron microscopy analysis showed that zinc sulfide formed as the dissolved zinc concentration decreased in anaerobic batch tests containing SRB [12]. This would have implications on the removal of dissolved metals, which contributes to the attenuation of AMD [7].

Although the rate of sulfate reduction is low compared to other sulfidic environments [79], our study found evidence of sulfate reduction comparable to other acidic environments [62]. However, higher rates of sulfate reduction have been previously found [80]. One explanation for the difference in the rates of sulfate reduction is that the SRB at our site may be reducing a different terminal electron acceptor [14,16,42,43,44]. Another explanation is that since AMD sites are high in iron concentration, sulfide may exist too briefly, as it quickly reacts with iron to reduce it, and thus escape our measurements based on evolving sulfide compounds.

### 4.4. Coupling Sulfur and Iron Cycles 

Iron and sulfur are abundant elements on earth and are particularly plentiful in AMD sites. Sulfate reduction was thought to occur after the pool of iron or organic substrate was depleted because oxidized iron is more energetically favorable as the terminal electron acceptor at neutral pH [48,49]. However, AMD sites have high concentrations of dissolved iron and sulfur; therefore, there may not be a significant preference or progression of terminal electron acceptors utilized.

Research on iron reduction under acidic conditions is less extensive compared to neutral pH environments [1,12,79,81,82]. Even fewer studies on iron reduction by sulfide under low pH exist. For example, in a study that tested the influences of pH and acetate on the ratio of iron and sulfate reduction, the pH of the bioreactors were 6.0–7.5, a pH range higher than normally found in AMD-impacted sites [45]. This indicates that the cycling of iron influenced by sulfide in acidic systems might be more significant than previously understood, given the higher concentrations of dissolved Fe^3+^ ions in these environments compared to marine or subsurface settings.

In our study, we observed that Fe^3+^ interacts with sulfide through several mechanisms. Ferric iron can be precipitated by sulfide, with greigite (Fe_2_Fe_3_S_4_)—a mineral containing ferric and ferrous iron along with sulfur—being detected in our XRD analysis. This mineral was found in both bioactive and abiotic samples, indicating that greigite can form in the presence of sulfide regardless or microbial activity. Additionally, our results show that ferric iron is reduced by sulfide, as evidenced by the date presented in Figure 2 and Figure 3. However, the formation of ferric sulfide (Fe_2_S_3_) is not detected in the XRD data, aligning with its known solubility in acidic media.

While we did not specifically examine microbial reduction of ferric iron by organisms such as *Acidithiobacillus* or *Ferroplasma*, the presence of greigite in both bioactive and abiotic samples suggests that this mineral can form independent of microbial activity. The comparison between bioactive and abiotic samples further reveals that additional mineral species with lower average oxidation states are produced bioactive conditions, suggesting that microbial processes may enhance iron reduction.

The reactivity of dissolved Fe^3+^ ions with sulfide in acidic environments, coupled with the regeneration of sulfate or elemental sulfur through reactions with Fe^3+^ ions or microbial respiration, highlights the complex interactions between sulfur and iron cycles in AMD sites [4,18,83]. This underscores the substantial role of sulfide in the iron cycle, particularly under acidic conditions.

## 5. Conclusions

In this study, we described an additional mechanism by which SRB promotes attenuation of AMD by reducing iron via biogenic sulfide and provide additional information on the biogeochemical cycling of iron and sulfur. The implication of this study is that iron reduction mediated by biogenic sulfide may be more significant than previously thought in acidic environments.

## Figures and Tables

**Figure 1 microorganisms-12-01939-f001:**
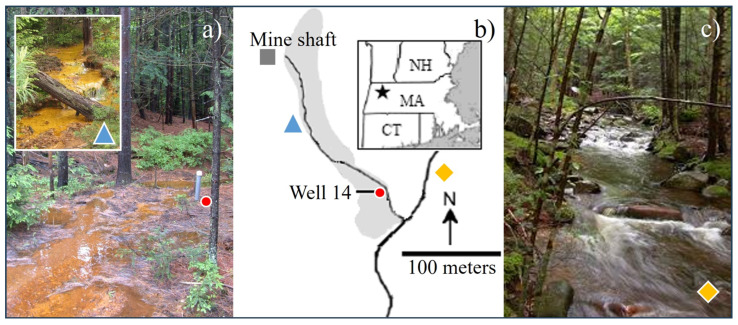
Davis Mine in Rowe, Massachusetts (black star) (**a**) effluent stream (blue triangle) from the Davis Mine shaft; (**b**) the left fork of the map shows the effluent stream from the mine shaft (gray square) past Well 14 (red circle) where all groundwater used in this study was collected, to the junction with (**c**) the Davis Mine Brook (yellow diamond). 42°40′51.3″ N 72°51′50.3″ W.

**Figure 2 microorganisms-12-01939-f002:**
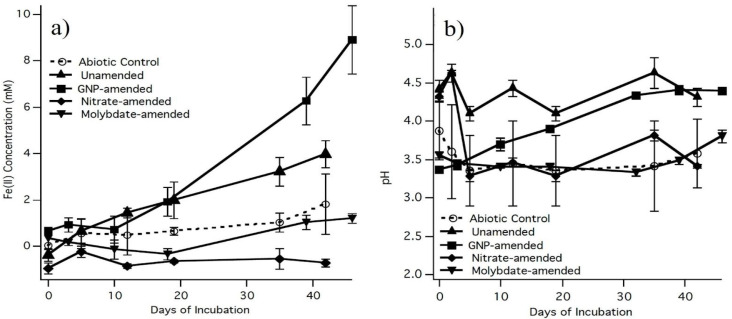
(**a**) Development of ferrous ion and (**b**) pH development of unamended (triangle), GNP-amended (square), nitrate-amended (diamond), and molybdate-amended (upside-down triangle) microcosms and the abiotic control (open circle) during the 42 days of incubation. Error bars indicate standard deviation of the mean of triplicate samples.

**Figure 3 microorganisms-12-01939-f003:**
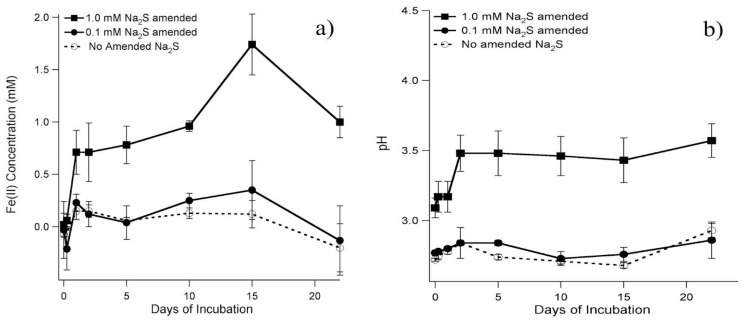
(**a**) Abiotic development of ferrous ion and (**b**) pH in abiotic microcosms with 1.0 mM Na_2_S (square), 0.1 mM Na_2_S (solid circle), or without Na_2_S (empty circle, dashed line). Microcosms were incubated for 22 days at a constant temperature of 20 °C. Error bars indicate standard deviation of the mean of triplicate samples.

**Figure 4 microorganisms-12-01939-f004:**
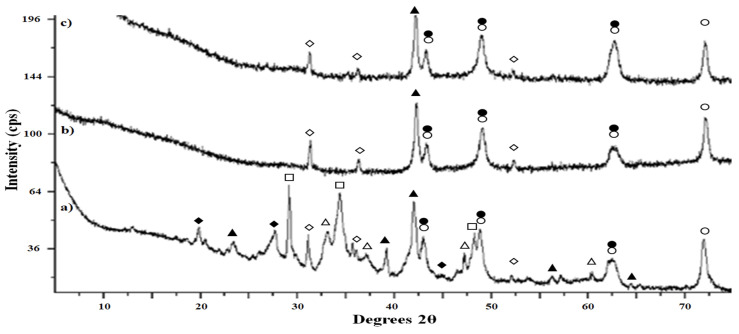
X-ray diffraction patterns of the black precipitates formed from in the soil-free Davis Mine water samples amended with 5 g/L Na_2_S. From the bottom to top: (a) bioactive, (b) abiotic with 5 g/L FeCl_3_, (c) abiotic with 5 g/L FeCl_2_. All samples display peaks consistent with hematite (open circles), magnetite (solid circles), and greigite (open diamonds), while the bioactive shows a marked increase lower valence minerals including troilite (solid diamonds), pyrite (open triangles), pseudomorphic pyrite (solid triangles), and chalcopyrite (open squares).

**Figure 5 microorganisms-12-01939-f005:**
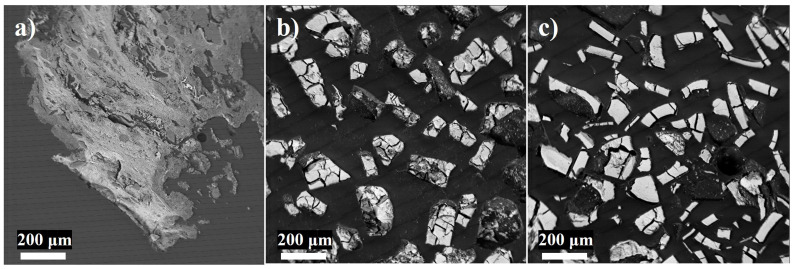
Electron microprobe imaging of black precipitates from (**a**) GNP-amended microcosms, (**b**) soil-free mine water FeCl_2_ + Na_2_S control, and (**c**) soil-free mine water FeCl_3_ + Na_2_S control.

**Table 1 microorganisms-12-01939-t001:** Sulfate reduction rate (SRR) experiment. All microcosms were in triplicates except for the abiotic controls, which were in duplicate. NA is not available and BDL is below detection level.

Microcosms	SRR (nmol/(cm^3^/d))	pH	ORP (mV)	DOC (mg/L)	Sulfate Conc. (mM)	Fe^2+^ (mM)
Unamended	19 ± 2	5.6 ± 0.1	43 ± 5	8.4 ± 0.5	5.1 ± 0.3	1.6 ± 0.1
Glycerol + N + P	22 ± 2	5.5 ± 0.3	−25 ± 9	13.6 ± 0.9	4.5 ± 0.1	1.0 ± 0.1
Abiotic control	0.3 ± 0.01	3.0 ± 0.1	375 ± 7	12.8 ± 0.35	NA	BDL
Groundwater	NA	5.3 ± 0.2	235 ± 32	NA	9.5 ± 0.8	BDL

**Table 2 microorganisms-12-01939-t002:** Quantitation of SRB by qPCR of the *dsr* gene and direct cell counts of unamended and GNP-amended microcosms.

Microcosms	SRB/mL (Initial)	SRB/mL (Final)	Cells/mL (Initial)	Cells/mL (Final)
Unamended	2.5 ± 0.8 × 10^4^	2.75 ± 0.01 × 10^6^	1.52 × 10^7^	1.89 × 10^7^
Glycerol + N + P	2.2 ± 0.7 × 10^4^	6.59 ± 0.06 × 10^6^	1.83 × 10^7^	2.84 × 10^7^

**Table 3 microorganisms-12-01939-t003:** Detection of six specific SRB groups in unamended and GNP-amended microcosms based on group-specific amplification of the 16S rRNA gene. Detected (+), not detected (−).

SRB Group	Unamended Microcosms	Glycerol + N + P Amended Microcosms
*Desulfotomaculum (dfm)*	+	+
*Desulfobacter (dsb)*	−	+
*Desulfococcus–Desulfonema–Desulfosarcina (dcc)*	−	−
*Desulfovibrio (dsv)*	+	+
*Desulfobulbus (dbb)*	+	+
*Desulfobacterium (dbm)*	−	−

**Table 4 microorganisms-12-01939-t004:** Elemental composition of black precipitates from unamended microcosms, Davis Mine on-site black precipitate, and iron chloride controls. Composition was determined by electron microprobe analysis and is presented as relative weight of Fe and S in percent weight (n = 10).

Precipitate Samples	Fe	S
GNP Microcosms	55.2 ± 18.0	39.0 ± 11.8
Davis Mine	64.9 ± 4.8	35.1 ± 4.8
FeCl_2_ + Na_2_S	47.5 ± 1.9	37.1 ± 1.4
FeCl_3_ + Na_2_S	45.5 ± 0.7	37.4 ± 1.0

## Data Availability

All data generated or analyzed during this study are included in this published manuscript.

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
