# Peer review of "Biogenic Sulfide-Mediated Iron Reduction at Low pH"

_microorganisms, 2024, doi:10.3390/microorganisms12101939_

Round 1

Reviewer 1 Report

Comments and Suggestions for Authors

This study investigates the natural attenuation mechanisms of AMD, with a focus on the role of biogenic sulfide in iron reduction. The research found that under acidic conditions, SRB reduce iron by producing sulfide rather than through enzymatic processes. Overall, this manuscript expands the understanding in the role of SRB in the natural attenuation of AMDd, suggesting that biogenic sulfide-mediated iron reduction may play a more significant role in acidic environments than previously thought, and reveals a link between the biogeochemical cycling of iron and sulfur. However, it still have some concerns should be addressed before publication.

Specific comments:

Line 44-46: The following paragraph explores the relevant mechanisms, so the sentence: “....., but it remains unknown which role bacteria play in this biogenic process' may be confusing.

Line 83: To provide readers with a more intuitive understanding, it is recommended to include a photo of the sampling environment in this section.

Line 138-146: Please provide references for this method.

Line 207-218: Please provide references for this method.

Line 251 (Fig. 1): Are there significant differences between the data of each treatment? Please conduct a statistical analysis so the results of the given assay will be well supported.

Line 271-272: Please provide references for this sentence.

Line 279-283: To present this part of the data more intuitively, it is recommended to add a curve plot.

Line 304: Please refer to the suggestions for Fig. 1.

Line 370: The font size of the superscript for “Fe2+” is not consistent.

Comments on the Quality of English Language

fine

Author Response

Comments 1: Line 44-46: The following paragraph explores the relevant mechanisms, so the sentence: “....., but it remains unknown which role bacteria play in this biogenic process' may be confusing.

Response 1: We have revised this section for clarity. See introduction end of second paragraph and beginning of third paragraph, Line 44- 50.

Comments 2: Line 83: To provide readers with a more intuitive understanding, it is recommended to include a photo of the sampling environment in this section.

Response 2: Agree. We now provide a map, photos, and coordinates as figure 1. Page 3, Lines 98-102.

Comments 3: Line 138-146: Please provide references for this method.

Response 3: We provide a reference that can now be found in page 4, 3rd paragraph, line 142.

Comments 4: Line 207-218: Please provide references for this method.

Response 4: We provide a reference on page 6, 1st paragraph, line 244.

Comments 5: Line 251 (Fig. 1): Are there significant differences between the data of each treatment? Please conduct a statistical analysis so the results of the given assay will be well supported.

Response 5: Agree. We have clarified the language around the statistical significance for both the ferrous ion and pH have been included in the additional data in this section. See results, 1st section, lines 273-276.

Comments 6: Line 271-272: Please provide references for this sentence.

Response 6: We provide a reference on page 7, section 3 of results, line 309.

Comments 7: Line 279-283: To present this part of the data more intuitively, it is recommended to add a curve plot.

Response 7: To present this part of the data more clearly, we added a statistical analysis to the results of Table 1 to show the change over the incubation period. See results, section 2, lines 319-321.

Comments 8: Line 304: Please refer to the suggestions for Fig. 1.

Response 8: We added statistical analysis to the results of Table 2.  See page 8, results, third section, lines 333-335.

Comments 9: Line 370: The font size of the superscript for “Fe2+” is not consistent.

Response 9: We have corrected the inconsistency. 

Reviewer 2 Report

Comments and Suggestions for Authors

This manuscript reports the use of sulfate-reducing bacteria to promote the acidic attenuation of acidic mine drainage by the mechanism of action related to iron reduction. In this study, the mechanism of sulfate removal from acidic mine effluents is elucidated and the transformation pathways are discussed in detail. The microbial community responsible for this process was analyzed, and the interactions between iron and sulfur cycles were thoroughly investigated. The manuscript was well-written, and the data were well-explained. Thus, I recommend the acceptance of this paper after minor corrections:

1. Please specify the novelty and advantages of the proposed system compared to the existing treatment.

2. Section 3.1: In the sentence, “SRB were stimulated with GNP, Fe²⁺ production doubled that of the unamended microcosms, increasing more than 8 mM over the 46 days of incubation,” please clarify if this increase is statistically significant and if it correlates with the sulfide concentration observed.

3. The three sets of samples used as controls did not contain the unamended microcosms, but subsequent experiments to determine the reduction rate of sulfate involved comparisons.

4. Figures: Fig. 1a showed the production of Fe²⁺ in microcosms with different treatments, while Fig. 1b only presented the pH changes over time. It is suggested that the authors provide additional figures or data that show the progression of sulfate reduction rates and the corresponding changes in pH and Fe²⁺ concentrations for better comparison.

5. Section 3.4:In the sentence, “The black precipitates that developed in the soil-free samples and the GNP-amended microcosms were analyzed using x-ray diffraction,” please clarify if these precipitates were the result of total internal resistance or if other factors were involved. The statistics provided in Table 4 should match those described in this section.

Comments on the Quality of English Language

There are a few details that need to be optimized in English.

Author Response

Comments 1: Please specify the novelty and advantages of the proposed system compared to the existing treatment.

Response 1: The interesting aspect in this study is the link between the biogeochemical cycling of iron and sulfur. Instead of relying only on iron reducing bacteria found in acidic environments, it may be more beneficial to enhance the activity of sulfate reducing bacteria instead.

Comments 2: Section 3.1: In the sentence, “SRB were stimulated with GNP, Fe²⁺ production doubled that of the unamended microcosms, increasing more than 8 mM over the 46 days of incubation,” please clarify if this increase is statistically significant and if it correlates with the sulfide concentration observed..

Response 2: We clarify the statistical significance as follows: Day 46 value for the unamended was 2.35 mM with a 95% confidence interval of 1.07, and the GNP was 8.89 mM with a 95% confidence interval of 3.36 giving a significant statistical difference in the two treatments. See Results section, first paragraph, lines 258-264.

Comments 3: 3. The three sets of samples used as controls did not contain the unamended microcosms, but subsequent experiments to determine the reduction rate of sulfate involved comparisons.

Response 3: Yes, there were three sets of microcosms (amended with GNP, nitrate, and molybdate) plus an unamended set and an abiotic set. The experiment to determine the reduction rate also had a set of unamended, amended with GNP, and a set of abiotic controls.

Comments 4: 4. Figures: Fig. 1a showed the production of Fe²⁺ in microcosms with different treatments, while Fig. 1b only presented the pH changes over time. It is suggested that the authors provide additional figures or data that show the progression of sulfate reduction rates and the corresponding changes in pH and Fe²⁺ concentrations for better comparison.

Response 4: Figure 1 will now be referred to as Figure 2 in the manuscript. Fig. 2a and Fig. 2b both show the same set of treatments of the same samples over the same day range. 2a shows the change in Fe(II) and 2b the pH. The change in sulfate rate is listed in the table as we cannot record the instantaneous rate of change of sulfate. The sulfate reduction rates were determined after 20 days of incubation of the microcosms. We did not measure the sulfate reduction rates in a progression, so we do not have the actual pH or Fe2+. However, the microcosms in figure 2 were prepared the same way as the microcosms used for the sulfate reduction rates experiment; therefore, still comparable.

Comments 5: Section 3.4:In the sentence, “The black precipitates that developed in the soil-free samples and the GNP-amended microcosms were analyzed using x-ray diffraction,” please clarify if these precipitates were the result of total internal resistance or if other factors were involved. The statistics provided in Table 4 should match those described in this section.

Response 5: The clarification that this set of samples were to determine the natural precipitation and effect of background redox reactions" has been made in the methods section in addition to clarifying the difference between the abiotic controls treated with NaS2 and the soil-free XRD samples. In the methods section, 3rd section, 5th paragraph, lines 139 - 150. Also, see Results, section 3.4, lines 355-358 for statistics.

Reviewer 3 Report

Comments and Suggestions for Authors

Comments and Suggestions for Authors
Acid mine drainage (AMD) is a very interesting object for microbiology and biogeochemistry. Particularly, this interest is based on the AMD role in: i) acidification of the environment, ii) metal pollution of the environment, iii) search and isolation of unknown acidophilic sulfate-reducing bacteria (SRB), and iv) SRB application for AMD remediation.

The authors studied their object, namely AMD in Davis Mine (Rowe, Massachusetts), more than 10 years:

- Becerra, C.A.; López-Luna, E.L.; Ergas, S.J.; Nüsslein, K. Microcosm-based study of the attenuation of an acid mine drainage impacted site through biological sulfate and iron reduction. Geomicrobiol J, 2009, 26, 9-20. https://doi.org/10.1080/01490450802599250

- Beccera, C.A. Role of sulfate-reducing bacteria in the attenuation of acid mine drainage through sulfate and iron reduction. Open Access Dissertation, 2010, 323. https://scholarworks.umass.edu/open_access_dissertations/323

The paper contains research data about a model AMD prepared in 2007 (! – Line 97) as a natural slurry mixed with the sterilized underground water. The research resulted in the following data:

- the AMD models contained SRB (shown with the DNA analyses);

- these SRB were viable (shown by analyses of the sulfate reduction rate with 35S-marker);

- supplementation of these AMD models with nutrient additives (glycerol + ammonium + phosphate) increased both sulfate reduction (= sulfide production) and iron reduction (= ferrous iron production);

- supplementation of the AMD models with nitrate or molybdate as alternative sources or inhibitors of sulfate reduction (some SRB can grow as nitrate-reducers, some SRB can use fermentation when sulfate-related enzymes are blocked with molybdate) decreased both sulfate reduction and iron reduction.  

According to these data, the authors suggested that ferric iron reduction in their AMD models was provided neither enzymatic by iron-reducing bacteria nor by SRB but with biogenic sulfide (Lines 12-13: “We hypothesized that biogenic sulfide-mediated iron reduction is one mechanism”).

Comments.

1. Goal of the presented study.

Goal of the presented research is unclear.

According to the postulates declared in the abstract (Lines 12-13), introduction (Lines 76-81), and conclusions (Lines 479-480), it seems that the main goal was to understand if ferric iron could be reduced with sulfide under specific experimental conditions of the model AMD (Line 106: the model AMD = “microcosm”).  

Through the whole study, the authors tried to confirm their hypothesis in indirect way via correlation of the iron reduction with sulfate reduction by comparing results in experiments with stimulated or suppressed sulfate reduction.

So, I ask authors to expand the study (the paper) with one simple experiment which can be carried out in a day. Please purge your AMD model (“microcosm”) with a gas flow that contains hydrogen sulfide (for example, nitrogen + injected hydrogen sulfide).

The scheme to get hydrogen sulfide is the same as you have already presented (Lines 166-168). Na2S can be used as an initial source for the hydrogen sulfide.

Then compare quantity of reduced iron in the purged and blank variants. This experiment shall test your hypothesis in direct way within a few hours.

2. The SRB diversity.

Lines 14-15, 283-287, Table 3.

Since 2007, there were some new SRB genera described. Their total number is over 70 (https://lpsn.dsmz.de/genus?page=D). Please rewrite the text about the SRB diversity more detailed and, as well, explain why you only limited the organic additive with glycerol while some authors used a mixture of glycerol and lactate for the acidophilic SRB (Kolmert, A.; Johnson, D.B. Remediation of acidic waste waters using immobilised, acidophilic sulfate-reducing bacteria. J Chem Technol Biotechnol, 2001, 76(8): 836 – 843. doi: 10.1002/jctb.453). This explanation will be helpful for the readers for understanding the presented different results for Desulfobacter and other groups (Lines 285-287, Table 3).

3. Direct enzymatic iron reduction by SRB.

As the authors have mentioned (Lines 71-72), “some SRB may also enzymatically reduce iron which could also contribute to AMD attenuation [14-16]”. It is correct. As well, I can present a few references more:

- Li, Y.-L.; Vali, H.; Sears, S.K.; Yang, J.; Deng, B.; Zhang, C.L. Iron reduction and alteration of nontronite NAu-2 by a sulfate-reducing bacterium. Geochim Cosmochim Acta, 2004, 68: 3251-3260.

- Ramamoorthy, S.; Sass, H.; Langner, H.; Schumann, P.; Kroppenstedt, R.M.; Spring, S.; Overmann, J.; Rosenzweig, R.F. Desulfosporosinus lacus sp. nov., a sulfate-reducing bacterium isolated from pristine freshwater lake sediments. IJSEM, 2006, 56(12): 2729-2736. 

- Ryzhmanova, Y.V.; Nepomnyashchaya, Y.N.; Abashina, T.N.; Ariskina, E.V.; Troshina, O.Y.; Vainshtein, M.B.; Shcherbakova, V.A. New sulfate-reducing bacteria isolated from Buryatian alkaline brackish lakes: description of Desulfonatronum buryatense sp. nov. Extremophiles, 2013, 17: 851–859. doi: 10.1007/s00792-013-0567-z.

So, this process is widely distributed.

According to the presented information, “A set of microcosms was amended with 5 mM molybdate to selectively inhibit SRB and another set was amended with 20 mM nitrate to competitively inhibit both SRB and FeRB” (Lines 114-116). These additives could change the SRB metabolism from iron reduction to nitrate reduction (with nitrate supplementation) or to anaerobic fermentation of some organic substrate (with molybdate supplementation). Thus, experiments with additives do not exclude direct enzymatic iron reduction by some iron-&-sulfate-reducing bacteria in the natural system.

Please supplement the discussion on this topic.

4. Some small corrections.

Line 13: “and tested this at an abandoned pyrite mine” – please change to “and tested this in experimental model systems”.  

Lines 52-55: “It is also  known that when sulfate-reducing bacteria (SRB) reduce sulfate to the gaseous form of sulfide, H2S, the result is a permanent decrease of total sulfate in the contaminated water.  At Davis Mine, we observed that sampled groundwater from wells smelled of H2S.” SRB reduce sulfate-ion to sulfide-ion. Under acidic conditions, this sulfide-ion forms hydrogen sulfide.

Author Response

(The authors gave the same response as above.)

Reviewer 4 Report

Comments and Suggestions for Authors

The following issues should be considered:

Line 93. Sample collection. Include the figure with a map of sampling site, as well as add coordinates in the text.

Line 109. 90 mL of autoclaved groundwater – include data on the parameters of the groundwater sample and content of metal ions and sulfates.

Line 113. GNP-amended. Explain abbreviature and add the amounts of substrates added.

Line 220. 3.1. Inhibition and stimulation of sulfate-reducing bacteria (SRB) in microcosms – Do you have the data on Fe3+ and sulfate-ions concentration change? Add the methods used for iron ion determination in the section 2.

Line 258. Add the data on pH, ORP, DOC.

Table 1. Add the data on initial parameters of groundwater used.

Line 267. “The average sulfate concentrations for the unamended and GNP-amended microcosms were both higher than their respective abiotic controls.” Explain the reason. Did you use groundwater sample with the same initial concentration of sulfate in all variant of experiment? Sulfate reduction should (in unamended and GNP-amended microcosms) led to the decrease in sulfate concentration, while in abiotic control, while in abiotic control, sulfate concentration should have remained the same.

Line 288. Table 3. Italicize genus names.

Table 4. According to data shown in the Table, precipitations were formed from both Fe2+ and Fe3+. In this regard, could you add the data on the dynamic of both Fe3+ and Fe2+ concentration during microcosm incubation?

Figure 3. Diagrams shown includes chalcopyrite peaks. Did your water sample contain copper ions? Did you control copper concertation?

Line 353. “However, in this study, we have determined that SRB, and not FeRB, are also responsible for iron reduction, which contributes to remediation by co-precipitation.” How did you exclude direct iron reduction? Some aerobic iron-oxidizers, which are present in AMD samples may shift metabolism to reduce iron ions oxidizing organic compounds and reduced sulfur compounds.

Line 458. Coupling Sulfur and Iron Cycles. Under your experimental conditions, ferric iron may interact with sulfide via 3 ways:

- ferric iron may be precipitated with sulfide;

- ferric iron may be reduced with sulfide;

- ferric iron may be reduced by microorganisms oxidizing sulfide/DOC (for example, Acidithiobacillus, Ferroplasma, etc.).

Consider all possible variants.

Author Response

Comments 1: Line 93. Sample collection. Include the figure with a map of sampling site, as well as add coordinates in the text.

Response 1: Agreed. We now provide a map, photos, and coordinates as figure 1. See Materials and Methods section, end of first paragraph, line 99-110.

Comments 2: Line 109. 90 mL of autoclaved groundwater – include data on the parameters of the groundwater sample and content of metal ions and sulfates.

Response 1: We included the initial parameters of the groundwater to include ions. See materials and methods, 4th subsection, third paragraph, line 183-190.

Comments 3: Line 113. GNP-amended. Explain abbreviature and add the amounts of substrates added.

Response 3: We clarified this point in the Materials and Methods section, 2nd paragraph, lines 125-127.

Comments 4: Line 220. 3.1. Inhibition and stimulation of sulfate-reducing bacteria (SRB) in microcosms – Do you have the data on Fe3+ and sulfate-ions concentration change? Add the methods used for iron ion determination in the section 2.

Response 3: We added the methods for measuring Fe2+ in section 2. We do have sulfate ion concentrations over time by ion chromatography. However, in a closed system like the microcosms, we observed that the sulfate concentration is effectively constant within statistical deviation. We see this even in the abiotic controls. Therefore, we used radiolabeled sulfate to measure sulfate reduction instead. As for Fe3+, the ferrozine assay does not directly measure Fe3+.  See page 4, 2nd paragraph, lines 156-158.

Comments 5: Line 258. Add the data on pH, ORP, DOC..

Response 5:  We added the pH, ORP and DOC to Table 1. See page 7, table 1, lines 305-306.

Comments 6: Table 1. Add the data on initial parameters of groundwater used.

Response 6:  We added this information on page 4, 5th paragraph, lines 183-190.

Comments 7: Line 267. “The average sulfate concentrations for the unamended and GNP-amended microcosms were both higher than their respective abiotic controls.” Explain the reason. {1}Did you use groundwater sample with the same initial concentration of sulfate in all variant of experiment? {2}Sulfate reduction should (in unamended and GNP-amended microcosms) led to the decrease in sulfate concentration, while in abiotic control, while in abiotic control, sulfate concentration should have remained the same.

Response 7:  {1} Within sets of microcosoms the water was collected from the stock groundwater vessel at the same time and have the same sulfate concentration. However, various microcosm sets would have water used from the vessel on different days. All samples used the same stock groundwater from a single well source collected on a single day, but there may be slight variations in concentration from the stock water aging in the lab. {2} While the sulfate concentration would be expected to drop in a situation where sulfate is a relative limited reagent, if it is in great enough excess a statistically significant drop in concentration would not be detected, as in this case. Furthermore, given that the native groundwater does have a large reservoir of sulfate, the relative change in the concentration does not show much change as this can be treated as a pseudo-constant concentration of sulfate was not set up as a limiting reagent.

Comments 8: Line 288. Table 3. Italicize genus names.

Response 8:  We italicized the genus names.

Comments 9: Table 4. According to data shown in the Table, precipitations were formed from both Fe2+ and Fe3+. In this regard, could you add the data on the dynamic of both Fe3+ and Fe2+ concentration during microcosm incubation?

Response 9:  Microprobe data can only give elemental data and not information on the local oxidation state. That could be posssible with XPS but we do not have reasonalble access to that instrument. During incubation the Fe(II) data was determined directly with a colorimetric assay and the total iron was determined via the same colorimetric assay if measuring during the incubation of microcosms or with inductively coupled plasma if measuring stock groundwater. Though the technique to measure the total iron is useful to gain some understanding of the concentration it has a larger inherent error as compared to the assay for ferrous iron. The method of mesasurment for ferric iron is to find the difference between the total and the ferrous iron, yielding a less precise measure and one we did not feel comfortable publishing given the uncertainty in the data.

Comments 10: Figure 3. Diagrams shown includes chalcopyrite peaks. Did your water sample contain copper ions? Did you control copper concertation?

Response 10:  The ground water from well 14 was found to contain copper by inductively coupled plasma to a concentration that could explain the presence of chalcopyrite in the bioactive sample. The average ratio of iron to copper in the native water was about 460 iron ions to every copper ion. As shown in other studies of minerals found in attenuated water samples at the Davis Mine, chalcopyrite has offen been observed (Gramp, 2010). By finding it in the bioactive sample but not the abiotic soil-free samples would suggest that the low copper concentrations migh not lead to a stoichiometric co-precipitation of iron and copper species but can in the presence of SRB. See Results, section 3.4, lines 371-374.

Comments 11: Line 353. “However, in this study, we have determined that SRB, and not FeRB, are also responsible for iron reduction, which contributes to remediation by co-precipitation.” How did you exclude direct iron reduction? Some aerobic iron-oxidizers, which are present in AMD samples may shift metabolism to reduce iron ions oxidizing organic compounds and reduced sulfur compounds.

Response 11:  We understand direct iron reduction as not enzymatic or influenced by the biota in the sample. If we share that definition, then the lack of a significant change in ferrous iron concentration measured in the abiotic and inhibited microcosms would be the control, or at least, a background to show that any direct iron reduction was statistically insignificant under the test conditions during the incubation. Although we did not attempt to target iron oxidizers that can also reduce iron, we did not see any production of Fe(II) in the molybdate or nitrate-amended microcosms. For example, when comparing microcosms where SRB were inhibited versus the unamended microcosms, we see that no iron is reduced when SRB is inhibited. If iron oxidizers were reducing iron, the amount is below detection level.

Comments 12: Line 458. Coupling Sulfur and Iron Cycles. Under your experimental conditions, ferric iron may interact with sulfide via 3 ways: - ferric iron may be precipitated with sulfide; - ferric iron may be reduced with sulfide; - ferric iron may be reduced by microorganisms oxidizing sulfide/DOC (for example, Acidithiobacillus, Ferroplasma, etc.). Consider all possible variants.

Response 12:  Agree. Our paper demonstrates that ferric iron is reduced by sulfide and also that that ferric iron is precipitated. Greigite, which was detected in the XRD, has ferric, ferrous, and sulfur. Ferric iron being reduced by sulfide can be seen in Figure 2 and 3. Ferric iron is stablized in acidic media and fully ferric sulfide (Fe2S3) is soluble at lower pH. This is further supported as there is no mineral XRD data in the American Mineral and RRUFF databases for full ferric sulfide. The next most likely sulfide to form is greigite (Fe(II)Fe(III)2S4, which was seen in our XRD data in both the bioactive and abiotic samples. The presence in both abiotic samples demonstrates that greigite will naturally precipitate without the intervention of SRB, though we are not attempting to elucidate if partial reduction of ferric to ferrous iron by sulfide or other environmental agent is the cause of the "background or abiotic" precipitation. What the comparison of the XRD or the bioactive sample the two abiotic samples is that the large increase of additional mineral species that are produced and many identified as having a lower average oxidations state than the abiotic set.

Round 2

Reviewer 3 Report

Comments and Suggestions for Authors

I have not received any responses from the authors to my 3 comments (1. Goal of the presented study; 2. The SRB diversity; 3. Direct enzymatic iron reduction by SRB).

The responses (presented after the 1st round) to the comments relate to other questions and are probably responses to another reviewer (!).

At the same time, in accordance with the corrected new version of the article, it is clear that the authors have made a number of changes in accordance with my comments.

1. “an abandoned pyrite mine” has been replaced with “experimental model systems”.

2. Data on experiments on sterile sulfide exposure have been included.

3. One of the three references I recommended has been added (Li et al., 2004), etc.

At the same time, at least two serious points remain uncompleted or undiscussed, namely the following ones.

1. About the SRB diversity. Lines 344-348 and Table 3. It is clear that the addition of organic substrate increased the number of SRB and even the diversity of detected genera. However, the total number of the known SRB genera is over 70 (https://lpsn.dsmz.de/genus?page=D). Please explain why you have only limited the organic additive with glycerol but not with other additives which are available to a larger number of SRB genera. Please expand the discussion of the results obtained: it will be more helpful for readers than a simple stating the fact.

2. About direct enzymatic iron reduction by SRB. Lines 130-132: "A set of microcosms was amended with 5 mM molybdate to selectively inhibit SRB and another set was amended with 20 mM nitrate to competitively inhibit both SRB and FeRB." This statement is inaccurate: the molybdate ion is similar in configuration to the sulfate ion and thus it blocks the reducing enzyme. However, in this case, SRB can continue other processes besides sulfate reduction. Many SRBs are not obligate sulfate reducers and are able to use other electron acceptors. Thus, experiments with additives do not exclude direct enzymatic iron reduction by some iron-&-sulfate-reducing bacteria in the system. Please supplement the discussion on this topic.

Author Response

Dear Reviewer,

We apologize for our clerical error. Please know we did not intend to neglect your valuable feedback.  Comments from both reviews are addressed below. 

Comments 1: Goal of the presented study. Goal of the presented research is unclear. According to the postulates declared in the abstract (Lines 12-13), introduction (Lines 76-81), and conclusions (Lines 479-480), it seems that the main goal was to understand if ferric iron could be reduced with sulfide under specific experimental conditions of the model AMD (Line 106: the model AMD = “microcosm”). Through the whole study, the authors tried to confirm their hypothesis in indirect way via correlation of the iron reduction with sulfate reduction by comparing results in experiments with stimulated or suppressed sulfate reduction. So, I ask authors to expand the study (the paper) with one simple experiment which can be carried out in a day. Please purge your AMD model (“microcosm”) with a gas flow that contains hydrogen sulfide (for example, nitrogen + injected hydrogen sulfide). The scheme to get hydrogen sulfide is the same as you have already presented (Lines 166-168). Na2S can be used as an initial source for the hydrogen sulfide. Then compare quantity of reduced iron in the purged and blank variants. This experiment shall test your hypothesis in direct way within a few hours.

Response 1: Thank you for the enthusiasm and creativity in attempting to strengthen the scope of this work. Regarding your suggestion to use hydrogen sulfide; the point of this experiment is to show that in these environmental samples, Fe2+ is produced when sulfide is present. Therefore, we think using a soluble sulfide source, such as sodium sulfide, is more preferable because it's more quantifiable than having to account for Henry's law for the gaseous hydrogen sulfide. Also, unfortunately access to living samples is no longer possible. 

Comments 2 for Round 1 and Comments 1 for Round 2: The SRB diversity. Lines 14-15, 283-287, Table 3. Since 2007, there were some new SRB genera described. Their total number is over 70 (https://lpsn.dsmz.de/genus?page=D). Please rewrite the text about the SRB diversity more detailed and, as well, explain why you only limited the organic additive with glycerol while some authors used a mixture of glycerol and lactate for the acidophilic SRB (Kolmert, A.; Johnson, D.B. Remediation of acidic waste waters using immobilised, acidophilic sulfate-reducing bacteria. J Chem Technol Biotechnol, 2001, 76(8): 836 – 843. doi: 10.1002/jctb.453). This explanation will be helpful for the readers for understanding the presented different results for Desulfobacter and other groups (Lines 285-287, Table 3).

Response 2: Thank you for your questions.  The choice in organic substrate used in this study was based on previous Davis mine microcosms that tested simple carbon sources: acetate, ethanol, pyruvate, lactate, and glycerol (Masters thesis of J. Harrison 2005). We had even tried readily available, complex, organic substrates found at Davis mine such as the wood chips and algae (Becerra, et al. 2009) for a potential large-scale, on-site remediation, as well as composted cow-manure (Masters thesis of A. Gilmore 2011).  See page 11, 2nd paragraph, lines 423-429.

References Added:

  • Harrison, J. Ferric Iron and sulfate reduction in the attenuation of acid mine drainage: a microcosm study. Master’s Thesis, University of Massachusetts, Amherst, Amherst, 2005.
  • Gilmore, A. Attenuation of acid mine drainage enhanced by organic carbon and limestone addition: a process characterization. Master’s Thesis, University of Massachusetts, Amherst, Amherst, 2011.

Regarding SRB diversity, we elaborate in the discussion section on SRB (page 11, paragraph 4, lines 446-451). When we initiated the project, we used these primers as they were the latest at the time.  We tested for the presence of six major phylogenetic groups of SRB based on Daly 2000. While there are more than 6 groups of SRB, in the context of this AMD study, these have been found in AMD sites (57, 61, 12, 54, 62-64]. According to Anantharaman et al, they "identified 20 distinct phylum-level lineages (Table 1), 13 of which were not known to have dsr genes." Delta-proteobacteria is the largest with 34 genomes reported compared to the next phylum-level lineage, Nitrospirae at 19 genomes reported. In the context of this study, the 6 major groups that we investigated were of the Delta-proteobacteria and Gram-positive bacteria (Desulfotomaculum and Desulfosporosinus). Desulfotomaculum genus have been found in low pH environments (Rambabu et al 2020). Sánchez-Andrea et al. summarized genera at AMD sites which includes Desulfovibrio, Desulfobulbus, and Desulfobacter. See page 11, paragraph 5, lines 462-463.

 References Added:

  • Anantharaman et al (2018) Expanded diversity of microbial groups that shape the dissimilatory sulfur cycle. ISME 12: 1715-1728.
  • Rambabu, K., Banat, F., Pham, Q.M., Ho, S.H., Ren, N.Q., Show, P.L. (2020) Biological remediation of acid mine drainage: a review of past trends and current outlook. Environ. Sci Ecotechnol Mar 19; 2: 100024.
  • Sánchez-Andrea, I., Sanz, J.L., Bijmans, M.F.M, Stams, A.J.M. (2014) Sulfate reduction at low pH to remediate acid mine drainage. Journal of Hazardous Materials 269(30)98-109.

Comments 3 for Round 1 and Comments 2 for Round 2: Direct enzymatic iron reduction by SRB. As the authors have mentioned (Lines 71-72), “some SRB may also enzymatically reduce iron which could also contribute to AMD attenuation [14-16]”. It is correct. As well, I can present a few references more: - Li, Y.-L.; Vali, H.; Sears, S.K.; Yang, J.; Deng, B.; Zhang, C.L. Iron reduction and alteration of nontronite NAu-2 by a sulfate-reducing bacterium. Geochim Cosmochim Acta, 2004, 68: 3251-3260. - Ramamoorthy, S.; Sass, H.; Langner, H.; Schumann, P.; Kroppenstedt, R.M.; Spring, S.; Overmann, J.; Rosenzweig, R.F. Desulfosporosinus lacus sp. nov., a sulfate-reducing bacterium isolated from pristine freshwater lake sediments. IJSEM, 2006, 56(12): 2729-2736. - Ryzhmanova, Y.V.; Nepomnyashchaya, Y.N.; Abashina, T.N.; Ariskina, E.V.; Troshina, O.Y.; Vainshtein, M.B.; Shcherbakova, V.A. New sulfate-reducing bacteria isolated from Buryatian alkaline brackish lakes: description of Desulfonatronum buryatense sp. nov. Extremophiles, 2013, 17: 851–859. doi: 10.1007/s00792-013-0567-z. So, this process is widely distributed. According to the presented information, “A set of microcosms was amended with 5 mM molybdate to selectively inhibit SRB and another set was amended with 20 mM nitrate to competitively inhibit both SRB and FeRB” (Lines 114-116). These additives could change the SRB metabolism from iron reduction to nitrate reduction (with nitrate supplementation) or to anaerobic fermentation of some organic substrate (with molybdate supplementation). Thus, experiments with additives do not exclude direct enzymatic iron reduction by some iron-&-sulfate-reducing bacteria in the natural system. Please supplement the discussion on this topic..

Response 3: Thank you for the references. We have supplemented the discussion on SRB as also suggested by the above comment. See page 11, paragraph 4, lines 442-447 and page 11, paragraph 5, lines 458-459. We revised for clarity, "A set of microcosms was amended with 5 mM molybdate to selectively 

inhibit sulfate reduction in SRB and another set was amended with 20 mM nitrate to competitively inhibit both SRB for sulfate reduction and FeRB for iron reduction." See page 3, 2nd paragraph, lines 123-125.

Comments 4: Some small corrections. Line 13: “and tested this at an abandoned pyrite mine” – please change to “and tested this in experimental model systems”. Lines 52-55: “It is also known that when sulfate-reducing bacteria (SRB) reduce sulfate to the gaseous form of sulfide, H2S, the result is a permanent decrease of total sulfate in the contaminated water. At Davis Mine, we observed that sampled groundwater from wells smelled of H2S.” SRB reduce sulfate-ion to sulfide-ion. Under acidic conditions, this sulfide-ion forms hydrogen sulfide.

Response 4: Thank you! We have revised lines 13 in the abstract and in the introduction, 3rd paragraph, 53-54 as recommended.

Reviewer 4 Report

Comments and Suggestions for Authors

Authors have improved the manuscript according to the commentaries. Thus, it may be accepted after minor revisions. Please, include information from the responses 11 and 12 in the manuscript.

Author Response

Comments:  Authors have improved the manuscript according to the commentaries. Thus, it may be accepted after minor revisions. Please, include information from the responses 11 and 12 in the manuscript.

 We included information for comment #11 from page 10, last paragraph, line 409 to page 11, first paragraph, line 421.

We included information for comment #12 from page 13, first paragraph, line 523 to 549.